# Highly Sensitive and Selective Detection of L-Tryptophan by ECL Using Boron-Doped Diamond Electrodes [note 1]

**DOI:** 10.3390/s24113627

**Published:** 2024-06-04

**Authors:** Emmanuel Scorsone, Samuel Stewart, Matthieu Hamel

**Affiliations:** Université Paris-Saclay, CEA, List, F-91120 Palaiseau, France; samuel.stewart@cea.fr (S.S.);

**Keywords:** boron-doped diamond, electrochemiluminescence, L-tryptophan

## Abstract

L-tryptophan is an amino acid that is essential to the metabolism of humans. Therefore, there is a high interest for its detection in biological fluids including blood, urine, and saliva for medical studies, but also in food products. Towards this goal, we report on a new electrochemiluminescence (ECL) method for L-tryptophan detection involving the in situ production of hydrogen peroxide at the surface of boron-doped diamond (BDD) electrodes. We demonstrate that the ECL response efficiency is directly related to H_2_O_2_ production at the electrode surface and propose a mechanism for the ECL emission of L-tryptophan. After optimizing the analytical conditions, we show that the ECL response to L-tryptophan is directly linear with concentration in the range of 0.005 to 1 µM. We achieved a limit of detection of 0.4 nM and limit of quantification of 1.4 nM in phosphate buffer saline (PBS, pH 7.4). Good selectivity against other indolic compounds (serotonin, 3-methylindole, tryptamine, indole) potentially found in biological fluids was observed, thus making this approach highly promising for quantifying L-tryptophan in a broad range of aqueous matrices of interest.

## 1. Introduction

L-tryptophan is an amino acid that is essential to the metabolism of humans [1]. It is involved in the biosynthesis of proteins, but it also plays other important roles, along with its metabolites such as serotonin, in neurophysiological functions [2,3]. L-tryptophan can only be obtained through food intake, and the recommended daily dose for adults is estimated to be between 250 mg and 425 mg [4,5]. Low levels of the amino acid could be compensated for by the consumption of L-tryptophan-based dietary supplements as a possible strategy to help in the treatment of disorders such as depression, insomnia, obesity, or Parkinson disease [6,7]. High levels of L-tryptophan may also be harmful for the central nervous system and have been suggested to accelerate aging and neurodegeneration processes [8,9,10]. Therefore, there is a high interest in the detection of this important amino acid in biological fluids, including blood, urine, and saliva, in medical studies, in particular in the area of neuroscience and oncology [11,12,13]. A variety of analytical methods have been reported for the determination of L-tryptophan, which include for instance high-performance liquid chromatography (HPLC) [14,15,16] and colorimetric and spectrofluorometric methods [17,18,19,20]. These methods require expensive equipment and trained personnel and are time-consuming. Therefore, there is a need for alternative solutions that are low-cost, rapid, accurate and easy to use. In this context, a range of electrochemical sensors has been investigated with promising results [11,21,22,23,24,25]. Electrogenerated chemiluminescence (ECL) is also attractive here because it offers potentially higher sensitivity than voltammetric methods and good selectivity. ECL methods usually take one of two different approaches. In the first approach, L-tryptophan’s ability to enhance or quench the luminescence of certain metal complexes is used. For example, tryptophan has been utilized as a luminescence enhancer of the Ru(bpy)_3_^2+^ and KMnO_4_ system [26]. A very low limit of detection (LOD), around 10 nM, was achieved; however, selectivity was found to be poor in this case. In particular, an increase in luminescence observed from other amino acids like histamine were present. A similar approach was also used as an end HPLC detector in order to distinguish between the two different enantiomers (L-tryptophan and D-tryptophan) down to the picomolar level [27]. Moreover, the ECL quenching ability of tryptophan has also been extensively investigated, in particular its effect on iridium complexes [28,29,30]. This relies on indole being subjected to electrochemical oxidation, which reduces the generation of electrogenerated excited states and quenches ECL emission of the iridium complex. While this method has been shown to be more selective than the signal-enhancing technique, it was found to display a non-linear response, meaning that quantitative measurements may be limited. In the second approach, L-tryptophan may be detected using ECL through its direct oxidation into an excited intermediate state that luminesces itself. This phenomenon was firstly reported by Sakura et al., who found that tryptophan would luminesce when oxidized on a platinum electrode at 0.78 V in a 0.1 M NaOH solution and in the presence of a bromating agent [31]. However, this method required very high concentrations of tryptophan to be present (>1 µM), and so it was not deemed to be useful for routine analysis. This process has recently been refined, first by Chen et al., who found that upon the addition of hydrogen peroxide, both indole and tryptophan would luminesce in a basic solution (0.03 M NaOH and 0.1 M KCl) [32]. Using this method, they were able to achieve an LOD of both indole and tryptophan around 100 nM. Since then, further attempts to improve the sensitivity of this method have been investigated, most notably by Zholudov et al., who tested the use of another oxidizing coreactant, namely tetraphenylborate (TPB), with tryptophan in aqueous solutions on glassy carbon electrodes [33]. Although this method did work in near-neutral conditions, meaning that it was good for biological assays, ECL emission was rather weak, with an LOD of 300 nM. Thus, hydrogen peroxide has been demonstrated to be the best coreactant so far for assisting the electro-oxidation of tryptophan. Moreover, boron-doped diamond (BDD) electrodes are known for their exceptional electrochemical properties, including a wide potential window in aqueous media > 3 V, low double-layer capacitance, high stability and low adsorption properties [34,35,36,37,38]. Such electrodes have recently shown to be capable of opening new perspectives in ECL analysis [39]. In this paper, we investigate the feasibility and performance of electrogenerated hydrogen peroxide at the surface of BDD electrodes as a new efficient way to detect L-tryptophan in aqueous solutions without the need for adding further co-reactant to the analytical solution.

## 2. Materials and Methods

### 2.1. Chemicals

L-tryptophan, luminol, phosphate buffer saline (PBS) tablets, tryptamine, indole, serotonin, and 3-methylindole were all purchased from Sigma-Aldrich, Saint Quentin Fallavier, France and were of analytical grade. Deionized water was obtained from a Direct-Q UV 3 water purification system (Merck Millipore, Guyancourt, France).

### 2.2. Electrochemical Cell

A three-electrode system was used, including lab-grown boron-doped diamond (BDD) electrodes as both working and counter electrodes, and a platinum wire as pseudo-reference electrode. Polycristalline BDD was grown by plasma enhanced chemical vapor deposition (PE-CVD) on a highly conductive 4-inch <100> silicon wafer in a SekiDiamond AX6500 diamond growth reactor. The growth parameters were 1% methane in hydrogen at a pressure of 40 Torr and microwave power 3.5 kW. Trimethylboron was added to the gas phase as a source of boron dopant. The thickness of the resulting diamond film was ca. 800 nm with a doping level of 2 × 10^21^ boron atom.cm^−3^, as determined by SIMS measurements. A scanning electron microscopy (SEM) image of the surface of the BDD layer is shown in Appendix A. Electrodes were fabricated using a 10 × 10 mm^2^ (working electrode, WE) or 15 × 15 mm^2^ (counter electrode, CE) section of BDD. Electrical contact was taken through the silicon backside using copper tape. Moreover, electrodes from cut sheets of 0.128 mm foil gold and 1 mm foil glassy carbon (Sigma-Aldrich, Saint Quentin Fallavier, France) were fabricated for comparison with BDD.

### 2.3. ECL Measurements

The two electrodes (working and counter), along with a Pt quasi-reference wire electrode, were then placed inside a 10 mL custom three-dimensional (3D)-printed cell fitted with an optical glass window, so that the worker electrode was facing the optical detector. When measuring total light emission, electrochemical measurements were performed using a potentiostat Autolab PGSTAT128N and the photon emission was measured with a PDM03-9107-USB photomultiplier tube (ET Enterprises, Uxbridge, UK) using the following parameters: acquisition time 500 ms, high voltage 950 V. All measurements were carried out in the dark. Spectral data were obtained by placing the electrochemical cell into a Fluoromax 4P spectrofluorometer (Horiba Jobin Yvon, Paris, France). In this latter case, a portable PalmSens EMStat4 potentiostat was used for electrochemical excitation. ECL intensities were recorded at different fixed wavelengths with a slit size of 25 nm (i.e., ±12.5 nm). For example, point ‘350 nm’ was measured from 337.5 to 362.5 nm, and so on.

## 3. Results

### 3.1. In Situ Generation of Hydrogen Peroxide on BDD Electrode

BDD electrodes are particularity suitable for in situ generation of reactive oxygen species (ROS) in solution, due to their large potential window in aqueous solutions and low adsorption properties. ROS are a product of oxygen reduction reactions (ORRs) from dissolved oxygen and/or water, either through oxidation or reduction reactions. The simplest of these ROS is the superoxide anion radical, which is formed as dissolved oxygen gains an electron. This is often the first step towards the production of other molecules such as hydrogen peroxide [40,41,42]. The production of hydrogen peroxide on electrode surfaces may be assessed using the luminol molecular probe. ECL emission from luminol is enhanced in the presence of both the superoxide radical cation and H_2_O_2_, but at high cathodic overpotential ECL enhancement from H_2_O_2_ is predominant [43]. Figure 1 shows the evolution of ECL intensity of luminol on either gold, glassy carbon (GC), or BDD electrodes as cyclic voltammetry was performed starting from 0 V to a lower cathodic potential and then back to 1 V vs. Ag|AgCl, at a scan rate of 0.1 V.s^−1^. The lower potentials tested started from 0 V down to −3 V vs. Ag|AgCl in 0.5 V increments. As the resulting signals span several orders of magnitude, the results are displayed in a logarithmic scale. A typical voltammogram and resulting ECL signal recorded in such conditions is shown in Appendix A for the BDD electrode. When no reduction potential is applied (lower potential = 0 V), both the GC and Au electrodes perform better in the standard luminol/O_2_ system. As the lower potential is decreased to −1 V, there is a clear increase in ECL efficiency for all electrodes. This increase is less significant for Au electrodes, which start producing superoxide radicals at −1 V, where a reduction peak is observed on the voltammogram. However, the electrolysis of water occurs, producing hydrogen gas that will compete with the formation of ROS. The GC electrode features the largest increase in luminescence at this point. The generation of hydrogen peroxide seems to be maximal on GC electrodes from −1 V vs. Ag|AgCl. BDD has also a large increase in ECL emission, starting from −0.5 V vs. Ag|AgCl. However, as observed from ECL during the reduction reaction, the formation of H_2_O_2_ is not fully optimized until after −1.5 V due to the lower inner sphere reaction rate of BDD electrodes. When using a lower potential below −1 V for the Au and GC electrodes, steady ECL emission is reached as the generation of ROS is limited due to the high surface adsorption and the lower potential window in aqueous solution. The BDD electrode, however, continues to increase its luminescence as the lower voltage is decreased to the –3 V limit set, thus suggesting much higher efficiency of H_2_O_2_ generation on this electrode.

### 3.2. ECL Reaction in the Presence of L-Tryptophan

Simply oxidizing L-tryptophan on a BDD electrode in neutral conditions does not result in any luminescence, suggesting that the conditions that will produce an excited intermediate are not met. Thus, successive application of cathodic and then anodic polarization on BDD electrodes was examined in the presence of 500 nM tryptophan in 0.1 M PBS solution by cyclic voltammetry (Figure 2a). The appearance of an ECL signal is observed upon anodic polarization at approx. 0.9 V vs. Ag|AgCl. This occurs only after the reduction potential applied during CV reaches at least approximately −1.5 V vs. Ag|AgCl. According to previous experiments with luminol, this corresponds to the potential from which a large increase in the production of H_2_O_2_ occurs. This signal was found to gradually increase as a reduction potential was decreased further, down to −3.3 V vs. Ag|AgCl (Appendix A). However, potentials below −3 V were found to have significantly lower reproducibility between five replicate measurements. After this voltage, it can be assumed that other reactions, like the evolution of hydrogen gas at the cathode, disrupt the production of ROS that aid in enhancing the ECL signal. Similarly, the ECL signal was found to gradually increase while increasing the upper oxidation potential up to 1.7 V vs. Ag|AgCl. Above 1.5 V, reproducibility was also affected, hence an optimum value of 1.5 V vs. Ag|AgCl was considered. The effects of scan rate on the ECL signal emitted from L-tryptophan were also examined while scanning from 0 V to −3 V and then back to +1.5 V vs. Ag|AgCl in a solution of 100 nM L-tryptophan in 0.1 M PBS (Appendix A). Here it was found that ECL emission increased almost linearly from 10 to 250 mV.s^−1^. After ca. 300 mV.s^−1^, a plateau seemed to be reached, where the ECL signal intensity remains constant. Therefore, 250 mV.s^−1^ was found to be optimal. Taking into account the CV optimum values as determined above, a calibration curve was drawn in the range 0–1 µM L-tryptophan in 0.1 M PBS (Figure 2b). ECL measurements over 5 replicates demonstrated good reproducibility with the percentage relative standard deviation (%RSD) ranging from typically 3–9%. Calibration graph showed excellent linearity across the whole range (0–1000 nM) with a linear fit coefficient of determination of R^2^ = 0.99%. LOD (Blank + 3 × SD) and LOQ (Blank + 10 × SD) of 0.4 and 1.4 nM were achieved in PBS solution, respectively.

In order to establish the reaction mechanism and strengthen the hypothesis that the generation of hydrogen peroxide is responsible for the ECL of L-tryptophan, spectral data can provide valuable information. L-tryptophan is known to fluoresce at ca. 350 nm [44] when excited by UV light. However, all ECL measurements have shown maximum emission at different wavelengths, meaning that it generally goes through a chemical transformation before becoming ECL emitter. A spectrum with a peak at around 550 nm has been observed in the cases where a bromating agent is used as a coreactant. This is known to be associated with the chemiluminescence of the indole ring with the addition of a hydroperoxide group [45]. Moreover, L-tryptophan ECL can also produce luminescence at around 450 nm, which is said to be due to the formation of a dioxetane group [31]. In our study, ECL emission maximum intensity from L-tryptophan was recorded using the same electrochemical conditions as used for calibration in a spectrofluorimeter at different wavelengths in the range 350–700 nm with a slit size of 25 nm. The graph presented in Figure 3a is a reconstruction of the light emission recorded in these conditions according to wavelength. A maximum emission at ca. 425 nm is observed, suggesting that it is through the formation of the dioxetane intermediate that the ECL of L-tryptophan is generated. Dioxetane is common to ECL reactions and is known to be enhanced through the presence of ROS. Moreover, the same reaction was examined on the four different electrode materials (Au, Pt, GC and BDD) in 500 µM L-tryptophan solution. A range of lower potentials were tested (0, −0.5, −1, 1.5, −2, −2.5, −3 V) for each of the electrodes. It appeared that only the BDD electrode was able to generate an ECL signal from tryptophan in these conditions (Figure 3b). This unique ECL reaction to BDD suggests again that it is favored by the production of H_2_O_2_ produced at the surface of the diamond electrode and made available to ECL reaction through low adsorption at the electrode surface. Thus, a reaction mechanism is proposed in Figure 4. It is based partially on the work already performed by Sakura et al. and Chen et al., who examined the ECL reactions of tryptophan/H_2_O_2_/Br^−^ and tryptophan/H_2_O_2_ systems, respectively [31,32]. During electro-oxidation, H_2_O_2_ is generated through several steps of reduction and protonation through proton exchange with water. The tryptophan molecule firstly goes through electro-oxidation at 0.9 V and then is further oxidized through interaction with H_2_O_2_ to a hydroperoxide tryptophan intermediate. From this, a dioxetane group is formed, which is then protonated and rearranged to form the excited intermediate.

ECL presents the advantage of potentially being a very selective method, as only certain types of molecules are able to undergo this reaction. By examining the list of known ECL emitters, the most likely interfering species that occur naturally in human blood would be other polyaromatic compounds featuring specifically an indole ring. Four indolic compounds that have been reported to produce ECL emission under certain conditions were measured: (i) tryptamine, an indole ring with a 2-aminoethyl group substitution (ii) indole, the base indole ring, (iii) 3-methylindole, an indole ring with a methyl group substitution and finally a (iv) serotonin with a 2-aminoethyl and alcohol group substitution. 10 µM of each of the compounds were examined on BDD electrodes using the same conditions presented above. The ECL peak heights from each of the chemicals tested are presented in Figure 5. None of the compounds except for tryptophan gave a significant ECL signal over the blank.

## 4. Discussion

We have demonstrated that ECL emission from L-tryptophan can be achieved in aqueous solutions at the surface of BDD electrodes through successive reduction and oxidation processes. This unique reaction does not take place when other electrode materials such as gold or glassy carbon are used. Also, it appears to be highly specific to L-tryptophan in our experimental conditions among the few indole derivatives tested. The chemiluminescence of indole and some derivatives induced by electrogenerated superoxide ion on glassy carbon electrode in aprotic solvent was investigated by Okajima and co-workers [46]. They showed that superoxide ions produced at the electrode’s surface from dissolved oxygen could act as proton acceptors to indole compounds carrying hydrogen at the *N*-position. The so-formed hydroperoxy radical may proceed to the chemical oxidation of indole derivatives having an electron releasing group at the 3-position, leading to the formation of a 1,2-dioxetane intermediate through radical-radical coupling. A similar reaction has been observed in aqueous solution, e.g., on N-centered radicals derived from tryptophan at pH 10, when the latter radicals are produced by pulse radiolysis [47]. In this case, hydroperoxy radicals are produced from dissolved oxygen and water. However, the reaction described by Okajima and co-workers has been observed to be completely inhibited in acetonitrile upon the addition of water, quenching for instance the chemiluminescence of 3-methylindole [48]. This is thus explained here by the absence of indole radical formation in the presence of water (proton donor). Those observations suggest that the hydroperoxy radical could play a role in the chemiluminescence of tryptophan in water in some adequate pH conditions, given that an indole radical can be formed. This is supported by the fact that, according to Mahé and coworkers, hydroperoxy radicals are indeed mostly adsorbed on GC and Au surfaces and only made available in solution from BDD electrodes. Nevertheless, we have shown that at the high overpotentials used, hydrogen peroxide is the main species produced on BDD upon cathodic polarization. Therefore, we assume that H_2_O_2_ is the main contributor to tryptophan chemiluminescence in our study. When reversing polarization, an indole-derivative radical is formed on tryptophan and all interferences under test through electrochemical oxidation on the *N*-position [49]. Except for serotonin, all indole derivatives under testing have a similar structure but for the functional group in the 3-position. Hence, selectivity toward tryptophan can only be due to the nature of this group. At pH = 7.4, tryptophan is a zwitterion in which the amino group is protonated and the carboxylic acid is deprotonated. Thus we assume that, by contrast with the other interference species, the carboxylic group act as a proton acceptor, allowing interaction of hydrogen peroxide with the indole radical at the 3-position, leading to the dioxetane-like intermediate and subsequent excited state. We have investigated this reaction at physiological pH, which correspond to the pH of interest in many biological assays. However, the reaction is expected to be highly pH-dependent. A thorough investigation of ECL emission according to pH would certainly help with our understanding of the reaction mechanisms involved in the detection of tryptophan in water at BDD electrodes.

## 5. Conclusions

We have reported on a new electrochemiluminescence (ECL) method for L-tryptophan detection involving the in situ production of hydrogen peroxide at the surface of boron-doped diamond (BDD) electrodes. In analytical terms, this means that the reaction can be achieved without the need for an additional coreactant in solution. An ECL mechanism has been proposed based on spectroscopic data that involves the generation of a dioxetane group and subsequent excited state. A low limit of detection in the order of 0.4 nM has been observed, which to our knowledge is the lowest reported to date. Moreover, the detection has been shown to be rather selective against other indolic compound that may be found in particular in biological samples. Altogether, this makes this approach attractive for L-tryptophan detection in biological fluids or food products.

## Figures and Tables

**Figure 1 sensors-24-03627-f001:**
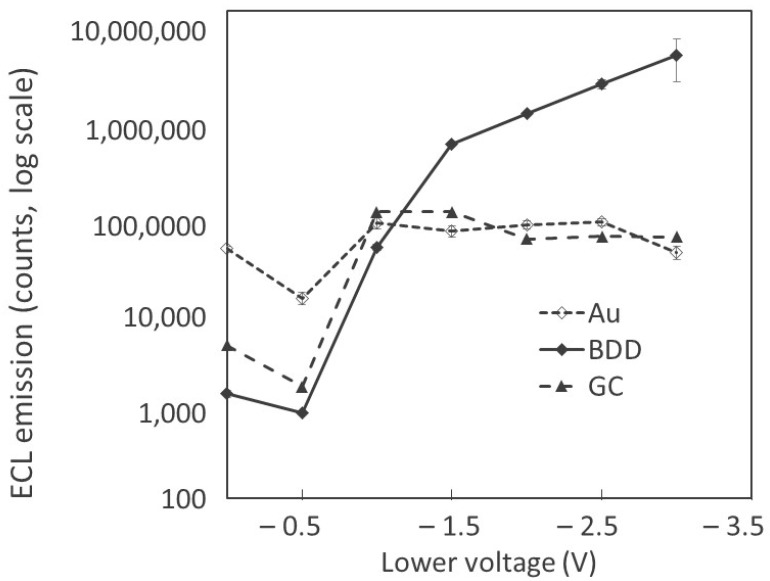
Lower applied voltage plotted against the corresponding oxidation ECL peak emission during the oxidation of luminol in log10 scale. Three electrodes, Au (short dash line) GC (long dash line), and BDD (continuous line), were tested, starting at 0 V before then sweeping to the lower potential and finally sweeping to 1 V, at a scan rate of 0.1 V.s^−1^ in a 0.1 M PBS solution with 1 µM of luminol (*n* = 3).

**Figure 2 sensors-24-03627-f002:**
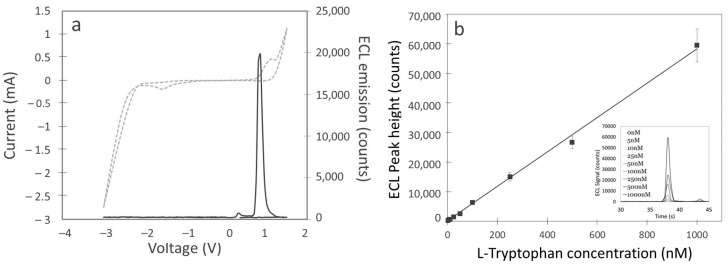
(**a**) CV (dashed line) and ECL (solid line) emission from 500 nM L-tryptophan in 0.1 M PBS solution. CV scan was 0 V → −3 V → 1.5 V → 0 V, at a scan rate 0.25 V.s^−1^. (**b**) Resulting calibration curve for L-tryptophan obtained in the same experimental conditions (inset: ECL peak emission for each concentration of tryptophan measured (0, 5, 10, 25, 50, 100, 250, 500, 1000 nM) in 0.1 M PBS (pH 7.4) on BDD electrodes after successive reduction and oxidation).

**Figure 3 sensors-24-03627-f003:**
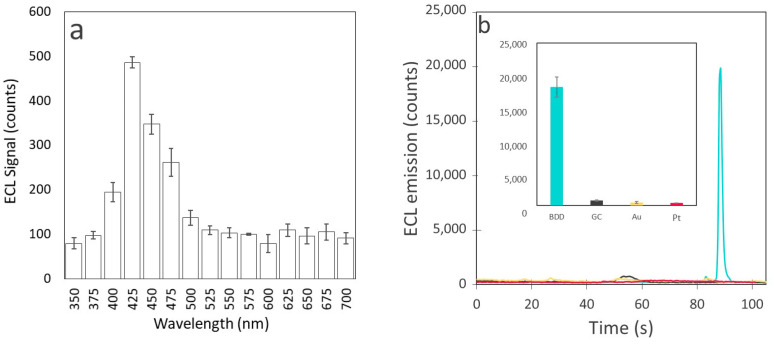
(**a**) ECL signal intensity versus wavelength from BDD electrode and (**b**) ECL emission plotted against time for four electrodes (yellow: Au, red: Pt, black: GC and turquoise: BDD) recorded with the PMT. In both cases a solution containing 0.1 M PBS (pH 7.4) and 500 nM tryptophan was used; CV scan was 0 V → −3 V → 1.5 V → 0 V, at a scan rate 0.1 V.s^−1^.

**Figure 4 sensors-24-03627-f004:**
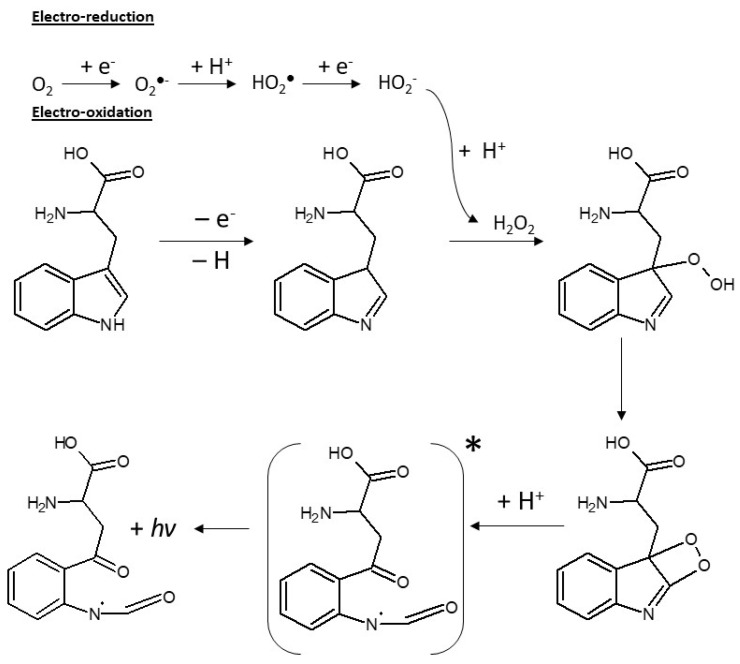
Proposed mechanism for the ECL of L-tryptophan at BDD electrode using successive electro-reduction and electro-oxidation.

**Figure 5 sensors-24-03627-f005:**
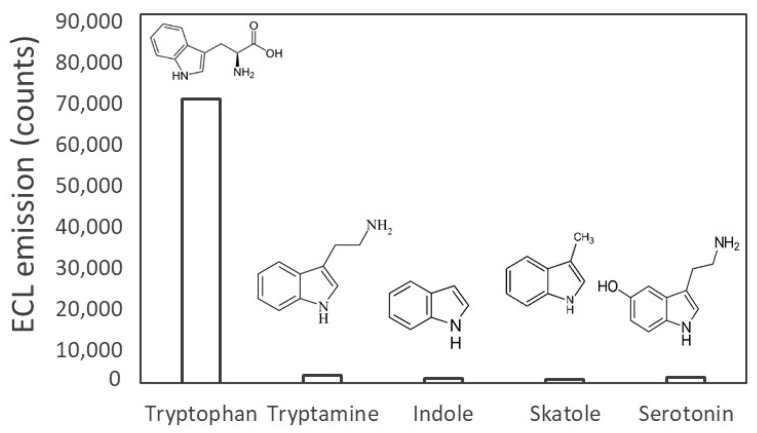
ECL maximum peak emission from 10 µM of five different indolic compounds (tryptophan, tryptamine, indole, skatole (3-methylindole) and serotonin) on BDD electrodes after successive reduction and oxidation. CV scan was 0 V → −3 V → 1.5 V → 0 V, at a scan rate 0.1 V.s^−1^.

## Data Availability

The datasets generated from the current study are available from the corresponding author on reasonable request.

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
