# Peer review of "Highly Sensitive and Selective Detection of L-Tryptophan by ECL Using Boron-Doped Diamond Electrodesâ€"

_sensors, 2024, doi:10.3390/s24113627_

Round 1
Reviewer 1 Report
Comments and Suggestions for Authors
Comments on manuscript sensors-3036273
Scorsone et al. presented an ECL-based method for selective and sensitive L-tryptophan sensing based on generating oxidizing intermediates at the interface of a boron-doped diamond electrode. Overall, the work is well-presented, and the obtained data are adequately discussed, supporting the main idea presented by the authors. The manuscript can be accepted after a minor revision provided below:
1. Introduction section, “it was found to display a non-linear response meaning that sensitivity is limited”. A non-linear response obscures quantitative measurement rather than impacting sensitivity alone.
2. “Materials and Methods” section, 2.2. Electrochemical cell, “A three-electrode system was used, including lab-grown boron-doped diamond (BDD) electrodes as both worker and counter electrodes, and a platinum wire as pseudo-reference electrode”. The electrode where the reaction of interest occurs is usually referred to as working electrode in the literature. Please edit the term “worker electrode”.
3. “Result” section, first line, edit in situ to in-situ.
4. “Result” section, line 111-112, “BDD electrodes are particularity suitable for in situ generation of reactive oxygen species (ROSs) in solution, due to their large potential window in aqueous solutions and low adsorption properties”. Please provide either relevant refences or actual data confirming the large potential window and low adsorption properties of BDD electrodes.
5. “They are a product of oxygen reduction reactions (ORRs)…”. Please specify “They”.
6. Figure 1., “Lower voltage used plotted against…”. Please edit to “Lower applied voltage”.
7. Figure 1., “Lower voltage used plotted against the corresponding oxidation ECL peak emission during the oxidation of luminol in log10 scale”. In the caption, the authors refer to the vertical axis as log10 scale, while in the actual figure it is shown as log scale. Please fix the discrepancy.
8. “3.2. ECL reaction in the presence of L-tryptophan” section, line 151-152, “Simply oxidising L-tryptophan on a BDD electrode in neutral conditions does not result in any luminescence”. More discussion and explanations for the possible reasons are needed.
9. Figure 2. (a), please specify the output corresponding to the dashed and solid lines.
10. Line 225, edit the typo “human blod”.
11. Line 233, “None of the compounds except for tryptophan gave a significant ECL signal over the blank”. A conclusive discussion providing explanations on possible reasons for the selectivity is critical for this part.
Comments on the Quality of English LanguageQuality of English only needs minor corrections, as indicated in Comments and Suggestions for Authors.
Reviewer 2 Report
Comments and Suggestions for Authors
This paper is worth publishing in Sensors, however, some revisions must be taken into consideration. The comments and corrections are highlighted directly in the text of the paper, as evidenced in the appended document.

Reviewer 3 Report
Comments and Suggestions for Authors
Overall a very nice body of work. It would be helpful to include an assessment of ECL sensors in biological matrices for context as well as the raw ECL produced by the different interferences as well as a control response so that the feasibility of this approach could be assessed. A comment on the pH dependance and how this is likely to affect the response should also be noted.
The discussion should be expanded and it would be nice to see a conclusion included
Comments on the Quality of English LanguageEnglish quality was very good with only very minor typographical errors.
